# Puparia Cleaning Techniques for Forensic and Archaeo-Funerary Studies

**DOI:** 10.3390/insects12020104

**Published:** 2021-01-26

**Authors:** Jennifer Pradelli, Fabiola Tuccia, Giorgia Giordani, Stefano Vanin

**Affiliations:** 1School of Applied Sciences, University of Huddersfield, Queensgate, Huddersfield HD1 3DH, UK; jennifer.pradelli@gmail.com (J.P.); tuccia.fabiola@gmail.com (F.T.); 2Dipartimento di Farmacia e Biotecnologie (FABIT), Alma Mater Studiorum Università di Bologna, 40126 Bologna, Italy; giorgia.giordani.gg@gmail.com; 3Dipartimento di Scienze della Terra dell’Ambiente e della Vita (DISTAV), Università di Genova, Corso Europa 26, 16132 Genova, Italy; 4National Research Council, Institute for the Study of Anthropic Impact and Sustainability in the Marine Environment (CNR-IAS), Via de Marini 6, 16149 Genova, Italy

**Keywords:** Diptera, identification, forensic entomology, funerary archaeoentomology

## Abstract

**Simple Summary:**

In forensic entomology, the correct identification of the species colonizing a body is fundamental. In old cases, puparia of Diptera represent the only entomological evidence available. Their identification is made particularly difficult not only because the lack of identification keys, but also because the presence on their surface of elements (dust, soil, dry decomposition fluids, bacteria, etc.) that can cover the diagnostic characters. Because of their fragility and the low amount of DNA, six cleaning techniques based on chemical and physical treatments have been tested. The results of this study indicate that cleaning via warm water/soap, the sonication and treatment with a sodium hydroxide solution are the best methods to achieve a good quality of the samples.

**Abstract:**

Diptera puparia may represent both in forensic and archaeo-funerary contexts the majority of the entomological evidence useful to reconstruct the peri and post-mortem events. Puparia identification is quite difficult due to the lack of identification keys and descriptions. In addition, external substances accumulated during the puparia permanence in the environment make the visualization of the few diagnostic characters difficult, resulting in a wrong identification. Six different techniques based on physical and chemical treatments have been tested for the removal of external substances from puparia to make identification at species level feasible. Furthermore, the effects of these methods on successful molecular analyses have also been tested as molecular identification is becoming an important tool to complement morphological identifications. The results of this study indicate that cleaning via warm water/soap, the sonication and treatment with a sodium hydroxide solution are the best methods to achieve a good quality of the samples.

## 1. Introduction

One of the most important taxa involved in the decomposition processes of animal organic matter is Diptera [1]. Flies, belonging to Calliphoridae, Sarcophagidae, and Muscidae families, are particularly important in legal investigation being the first colonisers of a body after death [2]. In medico-legal forensic entomology, the estimation of the minimum post-mortem interval (minPMI) and other evaluations about the relocation and/or concealing of a body, are possible only after an accurate identification at species level [3]. In fact, Diptera development, and more generally insect development, is temperature dependent but species specific, very often population related [4]. Moreover, the habitat preferendum, phenology, digging attitude, chronobiology, and distribution that represent the knowledge used to answer the investigative questions depend on the species [5,6,7,8].

Being associated with human decomposition Diptera are also commonly recovered from archaeological excavations. The study of insects associated with ancient human remains, such as natural and anthropogenic mummies or graves has been defined as Funerary Archaeoentomology [9]. This discipline provides information not only about thanatology (the scientific study of death and all the body modifications that happen after it), but also about funerary practices [10], season of death, social habits, and hygiene and health condition of past populations [11]. Also, in this case, any evaluation and interpretation of the archeological hypothesis needs a correct identification of the species [12].

Flies are holometabolous insect. Generally, the life cycle of flies includes egg, larval, pupal and adult stages. During the pupariation process, the larval cuticle goes through a series of chemical and physical changes, with the final formation of a hard case known as “puparium” [13]. It acts as a protective case in which metamorphosis takes place. After the adult emergence, the puparium is left empty on the site. Due to its high resistance to decay, puparia can be found in crime scenes [14,15], and they are particularly important in old cases when other developmental stages are no longer present. Moreover, these structures can be found even in archaeological contexts, where they might be the only traces of insect activity left after centuries or millennia [11,16,17,18].

The morphological identification of puparia is challenging due to the presence of a few diagnostic features on their outer surface. Most of the distinctive features are found in the posterior region, such as posterior spiracles and anal plate, and, on the ventral side of abdominal segment number 7, such as the size, shape, and distribution of spiculae [19]. It is worth mentioning that oral sclerites can be analysed from a puparium but often, especially in archaeological contexts, they are no longer present [2]. Puparia, depending on the context and on their conservation, are very often coated by external substances, like dust, decomposed fluids, dirt, fibres and soil debris which might cover and hide the above-mentioned diagnostic characters making difficult, if not impossible, a correct identification of the specimens [20].

During the past decades, several methods and techniques of insect cleaning, designed especially for adult beetles belonging to museum collections or for immature stages prepared for scanning electron microscopy (SEM) observation, have been described [21,22,23,24,25,26]. In literature, cleaning techniques are categorised into two main groups: methods based on a mechanical removal of the dirt particles, and methods based on a soaking system using different solvents [25]. The selection of which method is the most suitable for a specific specimen is strictly related to the state of insects’ conservation (how fragile the specimen is, the developmental stage considered, how old the sample is, etc.) and to the chemical and physical nature of the substance deemed to cover it. In principle, in order to be correctly identified, specimens have to preserve all the distinctive features after the cleaning treatment. Therefore, avoiding any damage to the sample is a priority. In practice, all methods and techniques affect the state of preservation of specimens, both molecularly and morphologically, although the extent of these effects can vary significantly based on the amount of time each sample is processed. Thus, it is important to balance the efficiency in processing entomological samples. In addition, because of the more and more common application of DNA techniques for species identification [27], also used for puparia identification [28], cleaning processes should not interfere with the DNA extractability and integrity.

In order to better understand which cleaning technique was the most suitable to be used for puparia, six chemical-physical methods have been tested. Two different experiments have been performed. The first one aimed to evaluate the efficiency of each method in removing external substances, improving the visual assessment of diagnostic features. The second one was designed to investigate the compatibility of each cleaning technique with potential molecular identification. Procedure guidelines are presented and tooltips for each method are listed.

## 2. Materials and Methods

Six methods were selected from the literature according to their ability to dissolve or remove specific substances (Table 1). Costs and availability of solutions were considered to select methods affordable by a standard laboratory. Diptera puparia in the families Calliphoridae, Sarcophagidae, Muscidae, and Sphaeroceridae, from forensic and several archaeological contexts were selected from the FLEA collection (Forensic Lab for Entomology and Archaeology based till 2019 at the University of Huddersfield (UK) and now at the University of Genoa (Italy)), and used for this study. All the specimens (ranging from 0.3 to 1 cm depending on the family) were visibly covered by substances of an unknown chemical composition deriving from the context of origin and therefore likely including non-insect organic material, botanical and soil residuals and other debris. As a result, the original appearance of the specimens was concealed. After a preliminary qualitative evaluation under a stereomicroscope (Leica M60, Leica, Wetzlar, Germany), the most adequate method according to the substances present was applied to the puparia (5–8 puparia tested for each cleaning method).

A pictorial archive of specimens before and after treatments was created using a Keyence VHX-S90BE digital microscope, equipped with Keyence VH-Z250R and VH-Z20R lens and VHX-2000 Ver.2.2.3.2 software (Keyence, Osaka, Japan).

(a)Warm water and soap solution: puparia were soaked in a solution of warm water (~60 °C) and commercial dish soap (depending on the brand of dish soap, component percentages might vary: sodium linear dodecyl benzene sulfonate, sodium lauryl alcohol triethoxy sulfate, lauric/myristic monoethanolamide, hydrotrope mixture, magnesium sulfate, colorant, petrolatum, perfume, ethyl alcohol 95%, deionized water) for approximately 10–30 min depending on the substances attached on their surface, and then they were wiped with paintbrushes. The processed samples were then rinsed with deionized water and air dried.(b)Sonication: puparia were placed separately inside vials filled with deionized water and then individually sonicated between 5 and 15 s, depending on the preservation status, using a sonicator bath (QH Kerry Ultrasonic Limited, *f* = 50 Hz). The samples were rinsed with clean deionized water and air dried.(c)Glacial acetic acid: puparia were gently wiped with a paintbrush soaked in glacial acetic acid (CH_3_COOH) or totally immersed in the acid for 5 min. They were then rinsed several times with deionized water, in order to stop the chemical reactions, and air dried afterwards.(d)Sodium hydroxide solution: puparia were immersed in sodium hydroxide (NaOH) 10% solution either for 5 or for 10 min. The solution was prepared by adding sodium hydroxide solid crystal to water. The samples were then washed gently in running deionized water to stop the chemical reaction, and air dried.(e)Hydrochloric acid/sodium bicarbonate: this method was described by Zangheri [30] in order to clean Coleoptera from museum collections. The method combines several different solutions in a pre-set order. Puparia were first immersed in distilled water for 24 h, and then placed in a clean vial with hydrochloric acid (HCl) for 10 min and soaked in a saturated solution of sodium bicarbonate (NaHCO_3_) for 15 min immediately afterwards. Finally, puparia were wiped with paintbrushes. The samples were washed with deionized water and air dried, prior to being microscopically observed.(f)Sodium hypochlorite: puparia were soaked for 5 and 10 min in a 5% solution of sodium hypochlorite (NaOCl). The specimens were washed under deionized running water and air dried before identification.

All the molecular analyses were performed on modern puparia of *Lucilia sericata* (Meigen, 1826) obtained from a breeding colony at the University of Huddersfield (UK). Puparia were subjected to two different treatments prior to DNA extraction. The first batch of puparia underwent previously described cleaning procedures immediately after the adults’ emergence; straight after DNA extractions had been performed (as a control, three puparia without any cleaning treatment were selected). The second batch, after adults’ emergence, was placed in small pierced plastic boxes containing a mixture of decontaminated horse blood, cat food, and ground soil, mimicking the conditions of thanatocoenosis and taphocoenosis. The containers were closed and stored inside the laboratory at room temperature. After seven days of incubation inside the mixture, the six cleaning techniques were applied to the puparia (as a control, six puparia were selected, three not placed in the mixture and not cleaned with any methods and three placed in the mixture but not cleaned with any techniques). In addition, further sodium hydroxide concentrations (saturated and 1%) were tested.

All DNA extractions were performed in triplicate on a single empty puparium using the QIAamp DNA Investigator Kit (QIAGEN, Redwood City, CA, USA). The manufacturer’s protocol was followed, and slightly modified by additional use of Proteinase K (100 μg/mL) from PROMEGA (Madison, WI, USA). Elution was performed with 50 μL of Buffer ATE. Quantification was performed using a Qubit^®^ 3.0 Fluorometer (Thermo Scientific, Waltham, MA, USA). Universal LCO-1490 Forward primer (5′ GGTCAACAAATCATAAAGATATTGG-3′) and HCO-2198 Reverse primer (5′-TAAACTTCAGGG TGACCAAAAAATCA-3′) were used [35,36] to amplify the mitochondrial COI gene (658 bp long) using Polymerase Chain Reaction (PCR). Master-mix reactions of 20 μL final volume were prepared following the PROMEGA GoTaq^®^ Flexi Polymerase protocol, which included Colourless GoTaq Flexi Buffer (5×), MgCl2 (25 mM), primers (IDT) (10 pmol/μL), Nucleotide Mix (10 mM), GoTaq DNA Polymerase (5 u/μL) and 2–4 μL of DNA template. The amplification programme (initial heat activation step at 95 °C for 10 min, 35 cycles of 95 °C for 1 min, 49.8 °C for 1 min, 72 °C for 1 min, and a final extension step at 72 °C for 10 min) was set up on a BioRad C1000 Thermal Cycler (Bio-Rad Laboratories, Inc., Hercules, CA, USA). A standard gel electrophoresis, in 1.5% agarose gel stained with Midori Green Advanced DNA Stain (Geneflow, Elmhurst, UK), was used to check PCR products. In case of positive results, 15 μL of PCR products were purified using QIAquick PCR Purification Kit^®^ (QIAGEN, Hilden, Germany) following the manufacturer’s instructions. Purified amplicons were sequenced by Eurofins (Eurofins Operon MWG, Ebersberg, Germany) following the standard Sanger method. For species identification purposes, DNA sequences were searched on GenBank database through BLASTn^®^ tool (NCBI, Bethesda, MD, USA).

## 3. Results

All the methods selected impacted the external appearances differently and led to various amplification results. Most of them removed the bulk of external substances after the first cleaning attempt. Examples of before and after-treatments are shown in Figure 1.

Despite the excellent visual results, it is worth mentioning that it is not always possible to achieve a totally cleaned puparium surface, due to the nature of the substances covering the puparia, which are a heterogeneous mixture. However, in the majority of the cases, specimens result cleaned enough to make visible the diagnostic features and allowing their identification. Different types of substances can simultaneously cover the external surface of a puparium. Hence, according to the composition of each substance, it may be necessary to use more than one method or to perform the same cleaning method several times to obtain a perfectly clean surface. Even though it may seem reasonable and fair to proceed until reaching the highest level of cleanliness, those multiple and/or combined methods affect the structure of the puparium. A brief qualitative analysis and tool tips for each method are listed below.

(a)Warm water and soap solution: This is the most affordable and the most effective method. The permanence of the puparia in warm water and soap can be prolonged as long as the operator is aware of the positive correlation between time and softness. This means that, during the final brushing, the operator needs to pay attention not to crush the puparium, which becomes more fragile.(b)Sonication: This method is particularly effective on encrusted debris. It works also on desiccated muddy or sludgy material, but in those cases, it is a time-consuming process. In fact, desiccated material, once rehydrated, usually stains the water inside the vial, not allowing a precise check on the status of the specimens treated. A multiple and/or prolonged sonication can widen the cracks present naturally on the puparia after the eclosion of the adults. In worst cases, posterior spiracles and anal plate can be ripped out from the puparium by the vibration, with the consequent loss of identification features. It is suggested, especially on archaeological samples, to check carefully the conservation status (presence of cracks on the surface) of each specimen prior to treat them with sonication. The more cracks are present, the less time the specimens should be kept in the sonication bath.(c)Glacial acetic acid: This method is effective at dissolving inorganic crystals. Commonly used by coleopterists, it was not previously tested on dipterous puparia. Due to its corrosive nature, low quantities and several rinsing steps are suggested. Some archaeological samples have to be evaluated closely before using acetic acid. In specific cases, due to the process of per-mineralisation (fossilisation process, during which mineral deposit creates a cast of the organism), a total or partial substitution of the organic matter can happen to the pupae. In these cases, acetic acid can destroy the sample totally.(d)Sodium hydroxide solution: The solution is very effective on samples covered by organic substances such as putrefactive liquids. This method is also commonly used to diaphanise larvae for slide microscopy [37].(e)Hydrochloric acid/sodium bicarbonate: It is the most time-consuming method as it involves an initial 24-h immersion in water. It is also the least effective of all methods, usually leaving a thin residue layer behind. Hence, additional cleaning with one of the other methods is also required.(f)Sodium hypochlorite: It is a common chemical and easily present in an entomological laboratory. It is known to disinfect and to react with many natural pigments. However, the solution is not particularly effective as a cleaning solution and, as a minor result, it decolours the specimens.

In term of DNA extractability, the results of the first group of puparia treated with cleaning procedures immediately after the adults’ emergence are presented in Table 2. DNA was positively extracted from the controls, from all the samples that were immersed in sodium hydroxide 10% solution for 5 and 10 min, from sonicated samples, from samples washed with water and soap, and from samples brushed with glacial acetic acid. However, DNA extraction failed when samples were immersed in glacial acetic acid, in bleach for 5 and 10 min, and with samples treated with the combination of hydrochloric acid and sodium bicarbonate solutions.

Results of the second group of puparia, which were placed in a mixture of decontaminated horse blood, cat food, and ground soil for a week and then cleaned, are presented in Table 3. DNA was extracted from all the controls: the three control puparia who were not placed in the mixture and not cleaned with any methods (CNTRL1) gave good quality PCR amplification; in contrast, the three control puparia placed in the mixture but not cleaned with any techniques (CTRLN 2) did not show any amplification. PCR was successful from samples immersed for 5 and 10 min in sodium hydroxide solutions (1%, 10% and saturated), from sonication, and from the samples brushed with glacial acetic acid. One sample washed in warm water/soap solution, and one sample immersed in glacial acetic acid, also showed positive results. The rest of the puparia cleaned with warm water/soap solution, immersed in glacial acetic acid and bleach, and treated with hydrochloric acid/sodium bicarbonate solutions did not show any positive results.

The mean quantification of both puparia groups was quite low, in agreement with the scarce availability of tissue in a single puparium suitable to extract nucleic acids from. In fact, most of the samples showed concentrations below the detectable threshold of the Qubit 3.0 fluorometer (0.001 ng/µL). However, DNA was successfully amplified, and the fragments successfully sequenced allowing to confirm the identification of all the specimens as *Lucilia sericata.*

## 4. Conclusions

All six methods selected successfully cleaned the puparia. However, if morphological and molecular analyses are taken into account together, the best methods, with positive results in both analyses, were the warm water/soap, sonication and sodium hydroxide solutions. Hydrochloric acid/sodium bicarbonate solutions, bleach, and glacial acetic acid immersion are, therefore, not recommended to clean entomological samples if molecular analysis is intended to be carried out. Furthermore, prolonged and multiple treatments with any of the cleaning methods might result in damage of insect remains and negatively affect DNA analysis.

## Figures and Tables

**Figure 1 insects-12-00104-f001:**
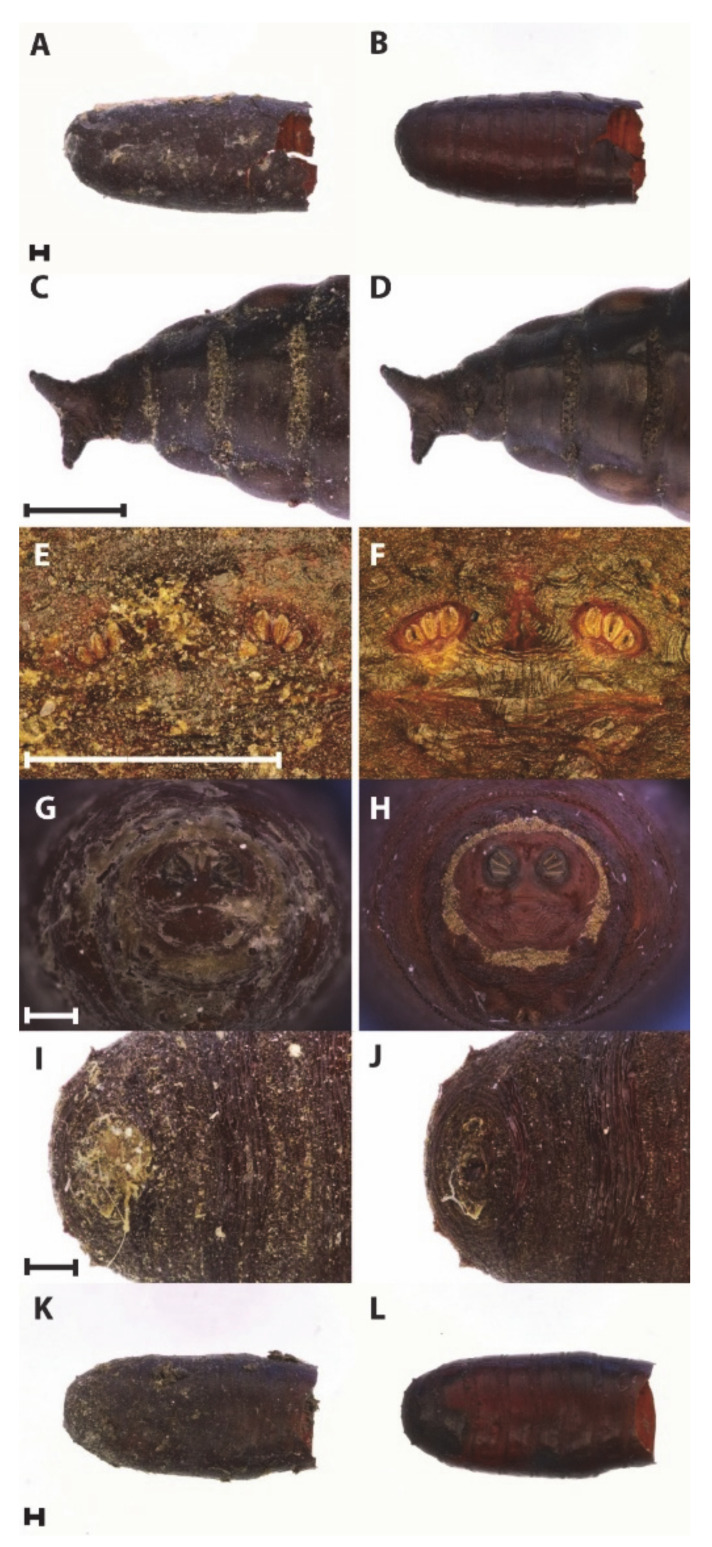
Puparia before and after treatment: (**A**,**B**) Water and soap solution, (**C**,**D**) Glacial Acetic Acid, (**E**,**F**) Sonication; (**G**,**H**) Sodium hydroxide 10% solution, (**I**,**J**) Hydrochloric acid/sodium bicarbonates solutions, (**K**,**L**) Sodium hypochlorite 1–5% solution. Scale bars: 500 µm.

**Table 1 insects-12-00104-t001:** Cleaning methods tested with reference and target substances.

Method	Suitable for	References
Warm Water and Soap solution (WH_2_O)	Fibres Dust Sludge	[16]
Sonication (SON)	Dross Soil debris Sand Botanical residues	[29]
Glacial Acetic Acid (GAA)	Inorganic crystals	[30]
Sodium Hydroxide solution(NaOH)	Putrefactive liquids Any organic matter	[31,32]
Hydrochloric Acid/Sodium Bicarbonate (ZAN)	Oily substances Grease	[30]
Sodium Hypochlorite (BL)	Organic matter Bacteria Mould/Fungi	[33,34]

**Table 2 insects-12-00104-t002:** Quantifications of DNA (average ± stdev) extracted from the first group of puparia cleaned immediately after adults’ emergence (CNTRL = control; NaOH = sodium hydroxide; SON = sonication; GAA = glacial acetic acid; WH_2_O = warm water and soap; ZAN = hydrochloric acid/sodium bicarbonate solutions; BL = bleach; ✓ positive results for the expected fragment, ✗ negative results for the expected fragment).

Samples	DNA (ng/µL)	PCR
CNTRL	0.427 ± 0.26	✓
NaOH 10%, 5′	<0.001	✓
NaOH 10%, 10′	<0.001	✓
SON	0.045 ± 0.034	✓
GAA immersed	0.408 ± 0.109	✗
GAA paintbrush	<0.001	✓
WH_2_O	<0.001	✓
ZAN	<0.001	✗
BL 5′	<0.001	✗
BL 10′	<0.001	✗

**Table 3 insects-12-00104-t003:** Quantifications of DNA (average ± stdev) extracted from the second group of puparia placed in the mixture for a week and then cleaned (CNTRL1 = puparia not placed in the mixture and not cleaned; CNTRL2 = puparia placed in the mixture and not cleaned; NaOH = sodium hydroxide; SON = sonication; GAA = glacial acetic acid; WH_2_O = water/soap; ZAN = hydrochloric acid/sodium bicarbonate solutions; BL = bleach; ✓ positive results for the expected fragment, ✗ negative results for the expected fragment).

Samples	DNA (ng/µL)	PCR
CNTRL_1	0.871 ± 0.219	✓
CNTRL_2	4.251 ± 1.988	✗
NaOH 10%, 5′	<0.001	✓
NaOH 10%, 10′	<0.001	✓
NaOH 1%, 5′	<0.001	✓
NaOH 1%, 10′	<0.001	✓
NaOH sat, 5′	<0.001	✓
NaOH sat, 10′	<0.001	✓
SON	0.053 ± 0.081	✓
GAA immersed	0.345 ± 0.373	✗
GAA paintbrush	<0.001	✓
WH_2_O	<0.001	✗
ZAN	<0.001	✗
BL 5′	<0.001	✗
BL 10′	<0.001	✗

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
