# Peer review of "Puparia Cleaning Techniques for Forensic and Archaeo-Funerary Studies"

_insects, 2021, doi:10.3390/insects12020104_

Round 1

Reviewer 1 Report

Nice experimental study.

1.

"In old cases, puparia, the pupal stages..."

→ Pls. refer to early approach: 

Ulf Grote, Mark Benecke (2001) Möglichkeiten der Zusammenarbeit von Forensischer Entomologie und Archäologie am Beispiel eines frühmittelalterlichen Gräberfeldes. In: Archäologisches Zellwerk. Festschrift für Helmut Roth zum 60. Geb., Hrsg. E. Pohl, U. Recker, C. Theune. Internationale Archäologie/Studia Honoria, Eds:: C. Dobiat, K. Leidorf, Publisher: Marie Leidorf Verlag, City = Rahden/Westf., Vol 16, p. 47 — 59

File here:

http://wiki2.benecke.com/images/7/75/Forensische_entomologie_archaeologie_grote_benecke_wesel.pdf

...and more recent:

Baumjohann K, Benecke M (2019) Insect Traces and the Mummies of Palermo — a Status Report. Entomologie heute, Vol 31, p. 73—93

File here:

https://www.researchgate.net/publication/340006039_Insect_Traces_and_the_Mummies_of_Palermo_-_a_Status_Report_Insektenspuren_und_die_Mumien_von_Palermo_-_ein_Statusbericht/link/5e726ef8299bf1571848a40a/download

2.

"...contexts were selected from the FLEA..."

→ Please mention where the lab is located (which university)

3.

"a)      Warm water and soap...."

"b)      Sonication...."

etc.

→ Please check with editor (I will, too) if the big distances between a) ... b) .... and the beginning of the text are okay.

4.

"QH. Kerry Ultrasonic..."

→ no dot after QH

5.

"All the molecular analyses were performed..."

→ Pls. mention which database you used to compare the DNA results against (Blast alone?)

6.

Does Insects recommend the use of

      (R)

for registered trademarks? Many journals do not use this. Pls. check (I will also ask editor).

7.

BLASTn(R)

→ dot at end of sentence

8.

"All the methods worked differently."

→ Maybe better: "...led to different amounts of DNA / ...different amplifications results" (something like that)

9.

"Soap"

→ which type / kind of soap?

Again, this is a helpful study.

Thank you. 

Author Response

In old cases, puparia, the pupal stages..."

→ Pls. refer to early approach:…

Reply: We thank the reviewer for this suggestion. In line 64 of the revised manuscript, we have added the new references  (14 & 15).

"...contexts were selected from the

FLEA..."

Please mention where the lab is located (which university)

Reply: Following the Reviewers’ suggestion, in lines 111 and 158 of the revised manuscript, we have added the physical location of the FLEA.

"a) Warm water and soap...."

"b) Sonication...."

etc.

→ Please check with editor (I will, too) if the big distances between a) ... b) .... and the beginning of the text are okay.

Reply: Following the Reviewer suggestion, we reduced the distance between the a), b), etc. and the beginning of the text.

We leave with the editor the final decision on the formatting.

"QH. Kerry Ultrasonic..."

→ no dot after QH

Reply: We removed the dot after QH at line 136 of the revised manuscript.

"All the molecular analyses were performed..."

→ Pls. mention which database you used to compare the DNA results against (Blast alone?)

Reply: Following the Reviewer comment, we mention the GenBank database (lines 190-193 of the revised manuscript).

Does Insects recommend the use of (R) for registered trademarks? Many journals do not use this. Pls.check (I will also ask editor).

Reply: We corrected the registered trademark symbol with ®.

We leave with the editor the final decision on the formatting.

BLASTn(R)

→ dot at end of sentence

Reply: We added the dot at the end of the sentence (line 193 of the revised manuscript).

"All the methods worked differently."

→ Maybe better: "...led to different amounts of DNA / ...differentamplifications results" (something like that)

Reply: Following the Reviewer comment, we rephrased the sentence as: “All the methods selected impacted the external appearances differently and led to various amplification results” (lines 197-198 of the revised manuscript).

"Soap"

→ which type / kind of soap?

Reply: We agree with the reviewer that this is an’important information.

In material & method section, lines 127-130 we described the kind of soap we used. “…commercial dish soap (depending on the brand of dish soap, component percentages might vary: sodium linear dodecyl benzene sulfonate, sodium lauryl alcohol triethoxy sulfate, lauric/myristic monoethanolamide, hydrotrope mixture, magnesium sulfate, col-orant, petrolatum, perfume, ethyl alcohol 95%, deionized water)”

Reviewer 2 Report

A nice little paper and helping hand for good methodology, but written down a bit hastily. A little more " love of detail" would be nice. The title e.g. includes the terms "forensic and archaeo-funerary" but there is literally nothing written in the introduction what is behind these terms and why exactly there is a need for ID of puparia. Interestingly there is some information in the abstract and the the simply summary and I hope this makes my request for a few more detailed explanations understandable for colleagues who are not familiar with these disciplines. There is still some space left in the introduction.

I do not understand reference [7] as a proof for "the more and more common application of DNA techniques for species identification, cleaning processes should not interfere with the DNA extractability and integrity". [7] is a book consisting of different chapters and I think it is good practise to refer to the accurate part in the book.

Which size were the puparia and how many samples were examined in total and by which method. The information here is scanty and inconsistent.

Where is FLEA located? The authors belong to three different institutions.

I suppose that "pupal cages of Diptera" should read "pupal cases" ?

Author Response

A nice little paper and helping hand for good methodology, but written down a bit hastily. A little more " love of detail" would be nice. The title e.g. includes the terms "forensic and archaeofunerary" but there is literally nothing written in the introduction what is behind these terms and why exactly there is a need for ID of puparia. Interestingly there is some information in the abstract and the the simply summary and I hope this makes my request for a few more detailed explanations understandable for colleagues who are not familiar with these disciplines. There is still some space left in the introduction.

Reply: We thank the reviewer for this helpful suggestion. Because it is of a great importance for us that our work results understandable for colleagues who are not familiar with these disciplines, we add some lines (39-77 of the revised manuscript) in the introduction in order to give a more complete background.

I do not understand reference [7] as a proof for "the more and more common application of DNA techniques for species identification, cleaning processes should not interfere with the DNA extractability and integrity". [7] is a book consisting of different chapters and I think it is good practise to refer to the accurate part in the book

Reply: We thank the reviewer for the comment.

We added the accurate chapter we refer to. New reference number is [27].

Which size were the puparia and how many samples were examined in total and by which method. The information here is scanty and inconsistent.

Reply: Following the Reviewer suggestion, we add some information in the material and method section.

For the morphological study, we worked on 5-8 puparia for each cleaning method (sentence added at lines 117-118 of the revised manuscript). Puparia were different in size based on the families the puparia belonged to. Following the Reviewer suggestion, we add a range of puparia size we used to make the reader more confident with this argument (line 112 of the revised manuscript).

For the molecular analysis we did all the DNA extractions in triplicate starting from a single empty puparium (line 171 of the revised manuscript)

Where is FLEA located? The authors belong to three different institutions.

Reply: Following the Reviewers’ suggestion, in lines 111 and 158 of the revised manuscript, we have added the physical location of the FLEA.

I suppose that "pupal cages of Diptera" should read "pupal cases" ?

Reply: Following the Reviewer comment, we rephrased the sentence as: “In old cases, puparia of Diptera…” line 17 of the revised manuscript.